# Reciprocally-Coupled Gating: Strange Loops in Bioenergetics, Genetics, and Catalysis

**DOI:** 10.3390/biom11020265

**Published:** 2021-02-11

**Authors:** Charles W. Carter, Peter R. Wills

**Affiliations:** 1Department of Biochemistry and Biophysics, University of North Carolina at Chapel Hill, Chapel Hill, NC 27599-7260, USA; 2Department of Physics and Te Ao Marama Centre for Fundamental Inquiry, University of Auckland, PB 92019, Auckland 1142, New Zealand; p.wills@auckland.ac.nz

**Keywords:** genetic coding, free energy transduction, non-equilibrium thermodynamics, transition-state stabilization, conformational change, aminoacyl-tRNA synthetases, catalytically active molten globules, emergent phenomena

## Abstract

Bioenergetics, genetic coding, and catalysis are all difficult to imagine emerging without pre-existing historical context. That context is often posed as a “Chicken and Egg” problem; its resolution is concisely described by de Grasse Tyson: “The egg was laid by a bird that was not a chicken”. The concision and generality of that answer furnish no details—only an appropriate framework from which to examine detailed paradigms that might illuminate paradoxes underlying these three life-defining biomolecular processes. We examine experimental aspects here of five examples that all conform to the same paradigm. In each example, a paradox is resolved by coupling “if, and only if” conditions for reciprocal transitions between levels, such that the consequent of the first test is the antecedent for the second. Each condition thus restricts fluxes through, or “gates” the other. Reciprocally-coupled gating, in which two gated processes constrain one another, is self-referential, hence maps onto the formal structure of “strange loops”. That mapping uncovers two different kinds of forces that may help unite the axioms underlying three phenomena that distinguish biology from chemistry. As a physical analog for Gödel’s logic, biomolecular strange-loops provide a natural metaphor around which to organize a large body of experimental data, linking biology to information, free energy, and the second law of thermodynamics.

## 1. Introduction

This review is motivated by a recent effort to define the physical forces leading to the origin of life [1]. It is a bold conjecture indeed that physicochemical forces drove production and selection of the biomolecules that enabled nature to (i) invent protein catalysts and heredity, (ii) store symbolic representations of nature in two distinct biopolymers, (iii) separate genotype from phenotype, and (iv) implement efficient mechanisms for capturing, storing, and transforming the chemical free energy necessary to sustain far from equilibrium states. We attempt here to build on that foundation by describing formally-coupled structures that arise from experimental studies of each of these processes and distilling shared characteristics that appear to define two novel kinds of forces, one constraining dissipative losses, the other driving innovation. These new forces incorporate and enhance the creativity potentiated by the distinction between hydrophobic and polar properties of matter described by Dill and Agozzino [1].

The examples are these:

(i) Coupling between ATP utilization and protein conformational change as Tryptophanyl-tRNA synthetase (TrpRS) activates tryptophan [2,3,4,5,6] creates an escapement or ratchet: Mg^2+^ ion accelerates catalysis if and only if the conformation changes, but the conformation change is thermodynamically favorable if and only if the resulting PPi product is released to solvent.

(ii) Studies of aminoacyl-tRNA synthetase (aaRS) evolution [7] led us to recognize that both protein folding and the genetic coding table depend intimately on amino acid side chain behavior [8,9,10]. Two sets of rules: folding—activity arises if and only if amino acid sequences fold—and coding—amino acid sequences arise if and only if the coding rules are obeyed—form a self-referential feedback loop accelerating the evolutionary search for polypeptides whose substrate recognition properties allow them to enforce the coding rules according to which they, themselves, were assembled [11,12].

(iii) AARS Urzymes—130-residue excerpts containing the most highly conserved modules from the full-length enzymes and which accelerate both amino acid activation and tRNA acylation—appear to be catalytically active molten globules [13]. Related work by Hilvert [14,15,16,17] demonstrated that catalysis occurs if and only if the catalytic structure of the molten globule assembles, and that conformation assembles if and only if the substrate is in its rare transition state configuration. Two rare species—folded Urzyme and reaction transition state—must thus occur simultaneously to achieve catalysis.

(iv) Expansion of the genetic code from a binary alphabet to the current 20 letters likely obeyed a variational principle minimizing dissipation of information and free energy, and aaRS evolution converted the former type of dissipation into the latter to enhance fidelity [18]. Errors could be reduced if and only if they became more energetically costly, but making errors energetically costly could be achieved if and only if errors were less frequent.

(v) The coding alphabet itself could expand if and only if there were enhancements in aaRS specificity; yet aaRS specificity could increase if and only if the alphabet size expanded [18].

Each example has been cast in identical format (Figure 1C) in which two distinct gated transitions are coupled by the fact that one filters, or gates the other. Our purpose here is to review these examples, which to our knowledge have never been considered as of a piece, and to examine why the associated logical operators function so powerfully.

## 2. Biological Process Control: Feedback, Autocatalysis, Hypercycles, and Strange Loops

Efforts to understand the origin of biology invariably invoke feedback. Even in its simplest form, feedback—autocatalysis—can be seen as a reflexive force by virtue of the fact that as a product increases its concentration induces change in its rate of formation, Figure 1A. Feedback also introduces a minimal historical context because the change in rates of synthesis connect present to past events. Kauffman [21] and Eigen [22,23] argued at about the same time (1971) that autocatalysis alone could not explain the origin of life, which, both argued, also required some form of integration, Figure 1B. Kauffman introduced linked “autocatalytic sets” [24]; Eigen introduced “hypercycles”, a closely related concept which, though conceptually similar, he advocated for different reasons having to do with limitations on the amount of information that could be retained in an error-ridden replicative regime [25,26].

An important recent contribution [19] derives a formal structure of autocatalysis from the matrix of stoichiometries, with reactions as columns and species as rows. Matrix representation provided a unified framework for distinguishing only five distinct patterns, which therefore compose a basis set for identifying autocatalytic subnetworks within arbitrarily large autocatalytic networks. That afforded a means to systematize the bewildering array of networks for modeling the emergence of life-like properties built by computational theorists as “Reflexive Autocatalytic Food Sets” (RAFS [27,28,29,30,31,32,33]) or GARDs [34], and show that they have the same formal structure [19]. The relevance of these control networks is almost certainly validated by their homologies to various chemical cycles, such as the formose cycle [35], the Krebs cycle [36], and related core metabolisms [37]. More recent studies strengthen arguments favoring metabolism first, autotrophic pre-biology forms.

There are compelling reasons to believe that additional, qualitatively different process control mechanisms may be necessary to account for the emergence of biology. The first of these is a pervasive, widely-recognized problem known variously as the error catastrophe [38], the “paradox of specificity” [39], or Eigen’s Cliff [40]. Turing [41] recognized the problem and advanced reaction/diffusion coupling as a solution as early as the mid-20th century. The phrase “paradox of specificity” actually refers alternately to opposite sides of a similar paradox. Szathmary [39] uses it in reference to the error catastrophe [38]: side reactions (parasites) develop, leading to extinction [26,38,42,43] because error-prone systems cannot function with the high specificity required for survival. An inverted guise of the problem, widely discussed by software and user experience (UX) designers [44], has a positive connotation. The rollaboard suitcase, designed with only flight crews in mind, unexpectedly appealed to a much larger market than products designed to meet needs of a broader clientele.

The qualitative term “error catastrophe” describes a dissipative process in which the proliferation of toxic variants or “parasites” overwhelms systems such as biological species in what has also been called “mutational meltdown” [42]. Experimental studies have verified and theoretical studies have qualified its existence. No general solution has been found for the toxicity of parasites [39]. Solutions invoked to rescue autocatalysis from error catastrophe include compartmentation [19] and reaction-diffusion coupling [41], but these are not general solutions. 

Reflexivity represents a second motivation to seek novel process control mechanisms. “Reflexive” in the context of autocatalysis refers to a simple closure property in which the product of the final catalyst provides the substrate for the first. Wills identified a fatal defect in autocatalytic sets, arguing that RAFS cannot, by themselves, account for the embedding of symbolic meaning into biomolecules [45,46,47,48,49] and hence that they cannot suffice to account for the emergence of biology. Carter and Wolfenden reinforced these arguments, noting that tRNA and mRNA sequences embed symbolic information about protein folding rules and which sequences fold sufficiently to function, respectively.

Examples (i–v) described in the Introduction share three formal properties that suggest a new class of process control structures with different and substantially more robust properties, Figure 1C. (i) They connect distinct realms (levels). (ii) Gating is represented by “if and only if” tests that reduce fluxes that pass from realm to realm. These complementary bi-conditional logical operators are logically equivalent to XNOR (Exclusive NOR; true if all inputs are true) gates in computer architecture. (iii) Coupling is effected by interchanging the respective antecedent and consequent of the two logical connectives. Interchanging creates a robust, but paradoxical, coupling we call “reciprocally-coupled gating”. The XNOT gates in this qualitatively different process control element address the error catastrophe directly, by restricting fluxes to those that fulfill the criteria.

Hofstadter [20] introduced the idea of strange loops to link such paradoxes, familiar from M. C. Escher’s intertwined staircases, to the Gödel incompleteness theorem. Extending self-reference from mathematics to material systems, he showed how the conditional—if and only if—often accompanies generation of paradoxes that can be resolved only by introducing meta-level axioms [See Chapter XV "Jumping Out of the System", pp 459–47420]. In the examples from experimental biochemistry and biophysics described in §3–§7, two such conditionals control one another. The resulting reciprocally-coupled gating becomes an especially powerful kind of self-reference that can generate ever-expanding novelty, in analogy to requiring new axioms. The logic of incompleteness may therefore underlie the generativity of non-equilibrium thermodynamic steady-states and other diverse biological phenomena.

## 3. Biogenetics: Free Energy Transduction Requires Reciprocally-Coupled Gating

Living things sustain themselves far from equilibrium by capturing the free energy of NTP hydrolysis and converting that free energy source efficiently into mechanical work, biosynthesis, and/or information. Detailed molecular mechanisms of that process, however, remained puzzling and incomplete for many decades [50,51,52], despite substantial theoretical advances [53,54,55,56,57]. It is generally accepted that molecular mechanisms which utilize that free energy are closely related to those necessary to convert the free energy of ion gradients into ATP synthesis, by which NTPs are then regenerated. We previously reported that catalysis of tryptophan activation by tryptophanyl-tRNA synthetase, TrpRS, requires relative domain motion to re-position the catalytic Mg^2+^ ion to accelerate ATP utilization, noting the analogy between that conditional hydrolysis of ATP and the escapement mechanism of a mechanical clock. The escapement allows the time-keeping mechanism to advance discretely, one gear at a time, if and only if the pendulum swings, thereby converting potential energy from the weight or spring driving the pendulum into rotation of the hands.

Conditional coupling of catalysis to domain motion, however, mimics only half of the escapement mechanism, suggesting that domain motion should be reciprocally coupled to catalysis by a complementary if and only if condition, completing the metaphor. Computational studies of the ligand-dependence of the free energy surface restraining domain motion later confirmed that reciprocal coupling: the catalytic domain motion is thermodynamically unfavorable unless the PPi product is released from the active site. These two conditional phenomena—demonstrated together only for the TrpRS mechanism—are ultimately driven by ATP hydrolysis, and function as reciprocally-coupled gates. The experimental data that underlie this novel allosteric mechanism (Figure 2) arose from attempts to understand the controversial question of how domain motion can contribute to catalysis, given that such motions are so much slower than transition-state formation and breakdown [58,59,60,61,62,63,64,65,66].

The answer to that question—that transition-state complementarity develops only transiently while the active site is being reconfigured by domain motion (Figure 2C)—emerged in pieces. (i) The first piece was to identify the core amino acid side chains that mediated the shear forces preventing domain motion [69], which identified the D1-switch residues, whose configuration changed most dramatically during catalysis (Figure 2A). (ii) Four of the seven side chains involved in that switching motif were then subjected to carefully designed combinatorial mutagenesis together with substitution of Mg^2+^ with Mn^2+^ as the catalytic divalent cation (Figure 2B) [5,73,74]. (iii) Complementary computational studies of the conformational transition by minimum action path analysis revealed that the mutated D1 switch residues, three of which are aromatic, change positions in the conformational transition state encountered during induced-fit, and that similar configurational changes of aromatic residues also defined the conformational transition states in unrelated dynamic proteins myosin and calmodulin [4]. (iv) A decisive new piece to the puzzle emerged when we found that single turnover kinetic measurements of the pre-steady state rate, k_chem_, for the combinatorial mutants exhibited the same pattern observed for k_cat_ with steady-state kinetics, and that k_chem_ (625/s) was itself actually slow enough to be comparable to timescales (ms) expected for domain motion [2]. (v) Finally, free energy surfaces computed by replica exchange discreet molecular dynamics validated the structure of the catalytic conformational transition state along the minimum action pathway [4] and showed that the conformational equilibrium shifted to favor the products conformation only after the PPi product was released (Figure. 2D) [75]. These separately demonstrated components compose the strange loop in Figure 2E.

As we and others [76,77] have noted, such an escapement mechanism is essential for efficient transduction of NTP hydrolysis free energy into other useful forms of mechanical or chemical work and/or information. Some implementation of both gating mechanisms—catalysis by domain motion and domain motion by catalysis—will thus likely be found in many other systems. In the present context, this observation provides an important clue regarding the general relevance of reciprocally-coupled gating to biology. By definition, efficiency means that a high proportion of the ATP hydrolysis free energy during amino acid activation is captured and employed to ensure aminoacylation of tRNA. Gating therefore greatly reduces the proportion of ATP that, from a system perspective, is wasted on dissipative, non-productive side reactions. 

## 4. Amino Acid Physical Chemistry Drove the Origin of Genetics

Dill and Agozzino [1] rightly attribute creative force to the physical properties of amino acid side chains and their behavior in water. Forces are gradients of energy, which in this context refers to changes in distribution constants between different environments, with respect to distance. From the work of Wolfenden [78,79,80,81], we can position each amino acid side chain precisely in a two-dimensional coordinate system whose axes are free energies of transfer from cyclohexane to water (polarity) and to the vapor phase (size). Moreover, those two free energies or proxies for them, are necessary and sufficient to estimate quantitatively the mean exposure to solvent of side chains in folded proteins, and to characterize the specific identity elements recognized by aaRS in cognate tRNAs [8,9,10].

We have argued [82] that genetic coding arose from an underlying duality in aaRS•tRNA cognate pairs first identified by Eriani [83]. That duality rests on impressive and comprehensive experimental data. (i) Primary [83] and tertiary [84,85,86,87] structural differences between Class I and II aaRS have been widely recognized as fundamental. (ii) The Class partitioning of amino acid substrates by contemporary aaRS appears to be according to their side-chain volume [8,9,10]. (iii) Class-dependent discrimination between cognate and non-cognate tRNA [88,89] and amino acid [90] substrates does not appear to depend on specific side-chains, but has been attributed to secondary structural differences between the two aaRS Classes. One can thus readily imagine quite deeply-based ancestry of the rudimentary distinctions necessary for the initial differentiation between coding letters.

It is nearly certain that the aaRS Class duality arose in the form of a bidirectional ancestral gene encoding an ancestral Class I aaRS on one strand and an ancestral Class II aaRS on the opposite strand [91,92]. (i) Experimental deconstruction of aaRS from both Classes confirmed that all essential catalytic activities required for genetic coding are retained in excerpts containing only the portions capable of antiparallel alignment of the corresponding coding sequences, and which have been called Urzymes on that basis [93,94,95,96,97]. (ii) Protozymes, representing~40% of the Urzyme sequences, have been encoded in a single bidirectional gene each of whose products accelerates amino acid activation 10^6^-fold [98]. (iii) Phylogenetic metrics derived from excerpts comparable to those examined experimentally differ by amounts that are statistically significant and which implicate earlier genetic origins for the Urzyme and protozyme excerpts [99]. Moreover, those metrics track linearly with catalytic proficiency. (iv) Antiparallel alignments of Class I and II middle codon-bases within the region of putative bidirectional coding retain statistically significant base pairing well in excess of that observed within Classes, and that frequency increases in antiparallel alignments of independently reconstructed ancestral nodes [100].

Making the enforcement of genetic coding rules (recognition of both cognate amino acids and cognate tRNAs) conditional on protein folding coordinated nature’s exploration of both protein folding and genetic coding rules (Figure 3A). As with the bioenergetic escapement mechanism, the antecedent and consequent levels for the two sets of rules are interchanged, and the rules themselves can thus be viewed as reciprocally-coupled gates governing which translated peptides fold and function, and which amino acid sequences are produced by translating messages (Figure 3B). We encounter here the depth of reflexivity sought by Wills [45,46,47,48,49]: the aaRS gene sequences whose translated products fold can, collectively, enforce the coding rules according to which they were assembled. Indeed, we have characterized the elements of Figure 3A as a minimal instruction set for launching the evolution of genetic coding, and by implication, as a “boot block” for installing Nature’s operating system [11].

Many details remain to be tested experimentally about this idea. (i) Can a bidirectional gene based on a two-letter alphabet produce active amino acid activation catalysts homologous to Class I and II protozymes [98]? (ii) Can Class I and II protozymes discriminate between two disjoint sets of amino acids? (iii) Can protozymes recognize and acylate cognate tRNAs with sufficient specificity? (iv) If not, can protozymes participate with suitable ribozymal catalysts to acylate cognate tRNAs [103]? How do protozyme specificities improve as the size of the coding alphabet increases? Notwithstanding, our previous work furnishes the experimental tools necessary to answer these questions.

## 5. Enzyme Catalysis Likely Evolved via Reciprocally-Coupled Gating

The work of Hecht [104,105,106] introduced the idea that catalysis may have been present in the incompletely-folded polypeptides likely to have dominated the population of molecules in the early proteome. Studies by Hilvert [16,17] confirmed that enzymatic activity does not require properly folded proteins, because a variant of chorismate mutase with identical catalytic activity to that of the native enzyme, is actually a molten globule. Experimental [15] and computational [107] studies of that system confirmed that the molten globular variant had a more negative entropy of activation than the wild type enzyme. That crucial result emphasizes that, once formed, the molten globular active site is actually a stronger catalyst than that of the native enzyme. It necessarily produces a more negative activation enthalpy to overcome the less favorable TΔS term to achieve the same catalytic proficiency. Remarkably, the intrinsically disordered molten globular structure of the monomeric chorismate mutase variant assumes a highly ordered tertiary structure in the presence of a transition-state analog inhibitor [17].

A key inference of these studies is that “(protein) folding can be coupled to catalysis with minimal energetic penalty” and that “many modern enzymes might have evolved from molten globule precursors.” [17]. Recent work in our lab showed that the TrpRS [13] and LeuRS Urzymes may also be catalytically active molten globules. Unpacking these remarkable statements, we discover a third form of reciprocally coupled gating. A chemical reaction catalyzed by a molten globule implies the simultaneous formation of two extremely rare species. The chemical transition state is, by consensus, an extremely rare species, and the low dispersion of the NMR HSQC spectra of a molten globule means that only a tiny fraction of molecules in that population have properly formed active-site configurations for catalytic activity. A simple re-formulation furnishes this description of what happens during catalysis by a molten globule: the concentration of the chemical transition state increases if and only if the active-site is properly folded, whereas that of the properly-folded active site population increases if and only if the substrate is in its rare, transition-state configuration (Figure 4A).

In this case, the experimental data—minimally dispersed HSQC spectra along the proton dimension, except in the presence of a transition state analog—provide a vivid demonstration of the paradox: the apparent transition-state dissociation constant, given by the rate enhancement, is many orders of magnitude greater than the ratio of transition-state to reactant concentrations, despite the fact that the concentration of properly configured active sites is also negligible for a molten globule in isolated solution. The apparent resolution to this paradox is that: (i) non-productive complexes between substrate and molten globule must exist at a much higher concentration than the E TS complex, and (ii) there must be important correlations between folding and catalysis by which the presence of the ground-state substrate can induce the ready formation of the catalytic configuration. That rationalization provides another clue to the relevance of the reciprocally-coupled gating (see §7).

Recent analysis reveals that even modern enzymes undergo something comparable to the dramatic coupling between folding and transition-state binding. As noted in S2, the CP1 insertion in the TrpRS Urzyme moves relative to the anticodon-binding domain. Because CP1 behaves extensively as a rigid body, it turns out that the locus of maximal frustration [109] along the polypeptide chain occurs in two places, where CP1 inserted into the Urzyme (Figure 4B). Remarkably, the local frustration at those two segments is progressively relaxed as the ground-state monomer in the open conformation (1MB2) binds ligands to form the PreTS state (1MAU), and then the transition state analog complex (2OV4). It re-emerges in the Products complex (1I6K). From the definition of frustration [108], this means that the ancestral junction between two different modules of the synthetase becomes most like a molten globule in the transition state complex. Hilvert’s thermodynamic analysis [15] implies that enhanced flexibility close to the conformational transition state allows the active site to wrap most tightly around and form stronger bonds to the transition state configuration of amino acid, adenosine monophosphate, Mg^3+^, and PP_i_ leaving group. 

## 6. Constraints on the Emergence of New aaRS tRNA Cognate Pairs

Expanding the genetic coding alphabet faced multiple, interrelated challenges (Figure 5). The first, recognized by Pauling on thermodynamic grounds [110], is that a single binding interaction cannot discriminate sufficiently between amino acids with similar side chains to assure high precision translation. Pauling’s observation irreducibly restricts the fidelity of codon-dependent protein synthesis accessible via equilibrium binding, ensuring that, absent specialized editing mechanisms, coded proteins would always remain quasispecies-like populations [111]. In the most difficult cases, notably valine, threonine, leucine, and isoleucine, the cognate aaRS must combine a second round of discrimination with hydrolytic editing to achieve the necessary quality control. An experimentally-based estimate of this limitation at the level of aaRS Urymes is given in Figure 4A of [18].

Hopfield [112] showed experimentally that high precision in such cases was achieved only by making mistakenly acylated tRNAs energetically costly. That transition converted the dissipation of information by translation errors by the less sophisticated ancestral aaRS into dissipation of free energy by the full-length aaRS (Figure 6A). The hydrolytic editing entails futile cycling of ATP to generate mis-acylated [113] tRNAs or less frequently mis-activated 5’-adenylates [114,115] that are then hydrolyzed. Mechanisms whereby aaRS gained hydrolytic editing capability were only the latest step in a progressive adaptation of aaRS Urzymes to insertion domains that enabled increased precision even for aaRS specific for amino acids that did not require hydrolytic editing.

The dimension of the genetic codon table imposes similar irreducible limits on the precision of tRNA aminoacylation by aaRS. For all coding alphabets of dimension smaller than 20, isoaccepting tRNAs may be acylated with multiple different amino acids, leading to a similar quasispecies-like distribution of coded proteins. Under these circumstances, aaRS within a single quasispecies will have slightly different specificities, and will randomly incorporate different percentages of related amino acid types.

A third challenge to the emergence of new aaRS•tRNA cognate pairs was that translation by ancestral aaRS and replication/transcription were necessarily coupled via the respective differential equations (Figure 6B). That coupling imposes a requirement for informational impedance matching between the respective error frequencies [18] in the two information transfer processes. Each reduced coding alphabet restricted errors to some currently unknown irreducible frequency, owing to limitations imposed by the quasi-species like distribution [111] resulting from the inability to select unique amino acids with high precision.

For these and other reasons, the evolution of the coding table remains a difficult and unsolved problem that will be solved only by constructing consistent joint evolutionary trees for aaRS tRNA cognate pairs. Delarue’s description of the descent of the genetic code as an asymmetric succession of binary choices [116] is perhaps the strongest extant model for the origin of codon assignments, because it provides the simplest explanation for the remaining redundancies. We have argued that the choice of pathways for the expansion of the coding table was not arbitrary, but formally resembled the shifting gears of a bicycle derailleur [11]. The succession of nodes in aaRS superfamily growth during expansion of the coding table from 2 to 20 amino acids is now potentially accessible by phylogenetic methods [99,117] and will supplement existing models for code development [116,118], perhaps in unexpected ways.

Each point of the triangle in Figure 5 can potentially be joined to the center by an antecedent => consequent relationship filtered by bi-conditionals. Errors made by aaRS can be reduced if and only if they become more thermodynamically costly, but errors can be made thermodynamically more costly if and only if errors are reduced. The number of distinct aaRS tRNA cognate pairs can be increased if and only if the coding redundancy is reduced, but the coding redundancy can be reduced if and only if the number of distinct aaRS tRNA cognate pairs increases. The dimension of the coding alphabet can be increased if and only if aaRS precision increases, but aaRS precision can be increased if and only if the coding alphabet dimension increases. We examine the latter example of reciprocally-coupled gating in detail in §7.

## 7. Reciprocally-Coupled Gating Shaped the Growth of the Genetic Code

Experimental decomposition of several Class I and II aaRS [93,94,95,96,119,120] defined three stages in aaRS evolution, which are summarized in Figure 6C. The initial production of both Class I (blue) and II (red) aaRS from bidirectional genes likely terminated before the coding alphabet had grown much beyond four distinguishable, but possibly overlapping amino acid types. The early strand-specialized era may have been initialized by the insertion of CP1 into Class I precursors, which inactivated bidirectional coding, and finished with the addition of anticodon-binding domains (ABD). Plausible partial speciation is indicated by the evolutionary trees in the center, and may have resulted in six different aaRS. Increasing the coding alphabet beyond a modest number of coding letters required evolving various kinds of specificity-determining mechanisms involving both energetic coupling between insertion and anticodon-binding domains and, eventually, specific editing domains.

Experimental data for Class I TrpRS showed that adding both insertion domains and ABDs enabled a final stage in which allosteric energetic coupling developed between the insertion and ABDs, leading eventually to development of high precision aminoacylation. That is likely to be true in outline also for other Class I aaRS and for the Class II superfamily. As noted along the bottom of Figure 7A these enhancements also required increased sophistication of protein folding. In turn, enhancing precision required including progressively more coding letters, which necessarily implies continual re-optimization of all or most proteomic sequences to exploit the advantages of the newly introduced distinctions between coding letters, as indicated by the cyclic bi-colored arrows in Figure 7A. 

Experimental evidence for identifying such coupling between aaRS specificity and the size of the coding alphabet includes the fact that the apparent limit to the alphabet size of which Urzymes (made from amino acids from the contemporary alphabet) are capable is roughly four letters: each of the Urzymes we have characterized can activate ~five of the twenty amino acids. As with speculations earlier in §4, this question is currently under investigation, as is the question of whether Urzymes can themselves be encoded using only a four-letter alphabet. That evidence appears to be confirmed by an increasingly steep ridge in the theoretical surface relating the impedance parameter (related directly to the informational transfer specificity) to the alphabet size and error frequencies (see Figure 4C in reference [18]). The preliminary specificity spectra of Class I and II Urzymes appears to sit at a privileged location on a low-energy passage over that surface.

A second line of experimental evidence stems from studies of the extensive allosteric communication by which TrpRS domain motion contributes to the amino acid specificity of full-length TrpRS [18,74,99,122]. Those studies complement work on amino acid discrimination in TrpRS and other aaRS [123,124,125,126] showing the difficulty of changing amino acid recognition just by mutating side chains that directly contact the amino acid itself.

The modular thermodynamic cycle comparing the TrpRS Urzyme with full-length TrpRS and the Urzyme plus either CP1 or the ABD [122] showed that domain motions increase specific discrimination of tryptophan vs closely related tyrosine. Combinatorial mutagenesis of the D1 switch confirmed that coordinated motion of D1 switch residues enhanced the rejection of tyrosine by ~4.4 kcal/mole above that exhibited by the Urzyme itself [74]. Neither intermediate modular construct improved specificity significantly. Exclusive dependence of enhanced aminoacylation [122] and specific side chain recognition by full-length TrpRS on interdomain coupling energies between the two accessory modules argues that independent recruitment of CP1 and the ABD during evolutionary development of Urzymes would have entailed significant losses of fitness. Development of high precision aminoacylation during aaRS evolution from the Urzyme stage to the full-length enzyme thus presents a paradox.

Notably, the editing domains present in subclass IA aaRS (ValRS, LeuRS, IleRS) are all outgrowths of the initial CP1 insertion. Achievement of the sophistication necessary for these functions was highly unlikely without increased numbers of coding letters. A final strange loop thus connects the genetic coding table to the proteome itself via the collective precision of the aaRS and the dimension of the genetic coding alphabet (Figure 7A; [18,99]). The idea that proteins evolved in distinct stages has found support in the multiple varieties of polypeptide architectures found in ribosomal protein structures [121], which are “frozen in time by virtue of their structural complementarity to ribosomal RNA structures”. Such variations must have been encoded into the sequences of mRNAs and genes concurrently with the dawn of heredity and natural selection [18,82,99,127].

Although this new perspective on the genesis of the coding table faces many challenges in order to validate and elaborate supporting details, we can outline how the pieces likely fitted together. Key elements of this strange loop are the evolving precision of ancestral aaRS and the dimension of the coding alphabet, which participate in the reciprocally coupled gating (Figure 7B) that guided the introduction of new aaRS tRNA cognate pairs together with other constraints illustrated in Figure 5 and discussed in §6. The emerging tree of new cognate pairs was shaped by the new possibilities introduced by enlarging the dimension of the coding alphabet.

## 8. Conclusions: Strange Loops Tame Eigen’s Cliff and the Paradox of Specificity

We return to the question raised by Dill [1] about “forces” that drive molecular self-organization and, ultimately, produce biology. A central tenet is that the experimental data summarized here furnish sufficient examples to warrant generalizing and arguing by analogy. We also acknowledge the insights of Deacon [128] and Russell and Branscome [76,77], both of whom recognized the relevance of mutual coupling via reciprocal linkage to the problem of sustaining far from equilibrium states as well as the importance of absence: “…use comes from what is not there” [129].

Process control elements regulate events in time. Much effort has been devoted to trying to understand biology in terms of a single process control element, feedback, via autocatalysis and/or hypercycles—a term reminiscent of introducing increasingly sophisticated “epicycles” to rescue the Ptolemaic model for the solar system. Those efforts encounter two stumbling blocks: (i) no intrinsic filter exists to limit toxic impacts of dissipative loss (parasites), and (ii) autocatalysis has only the second law of thermodynamics to define temporal direction. The examples outlined here identify a related, but more robust process control element whose novel features appear to surmount the limitations on autocatalysis arising from Eigen’s paradox and the paradox of specificity. Moreover, they operate on timescales from sub-millisecond to aeons.

The coupling together of two XNOR gates by interchanging the antecedent and consequent logical elements is a recurring formal description for many of biology’s most interesting and challenging questions. It is of interest to ask what else these occurrences have in common by collecting the clues we have identified along the way. Figure 8 is drawn to emphasize that reciprocally-coupled gating functions like a compound logical computer gate. Similarities and differences between Figure 8 and Figure 5.8 of reference [129] should be noted. The latter incorporates reciprocal coupling, but lacks the logical connectives that constrain dissipation. We note below how the coupling in Figure 8 could be described as “teleodynamic” because of how it creates the pretense of purpose.

The properties of reciprocally coupled gating are two-dimensional, exhibiting different properties—efficiency and incompleteness—in the vertical and horizontal directions of Figure 8, respectively. We argue that these properties are functionally analogous to potentials, so that their gradients correspond, respectively, to gravitational and chemical potential “forces”. Further, these two forces are associated, respectively, with the biological properties of survival and innovation.

*Efficiency and survival*. The vertical direction in Figure 8 corresponds to filtering each stream by the other, minimizing dissipation. The two XNOR gates eliminate parasites by restricting fluxes that do not obey the” if and only if” criteria. We liken this to gravity because gravity’s centripetal force restricts planetary motion to ellipses (consider here Feynman’s integral over paths formalism [130]). The strange loop in bioenergetics (Figure 2) ensures that NTP-dependent free energy transduction processes in biomechanical work, biosynthesis, signaling, and ATP synthesis are highly efficient. Variants of each type of antecedent—conformational state and transition-state binding affinity—are excluded unless they both meet the coupled logical bi-conditional filters.

Efficiency is also a hallmark of the remaining examples, most evidently in Figure 4 and Figure 7. The efficiency associated with the strange loop in Figure 3 is subtler, but no less important. We noted in [11] that the feedback loops illustrated in Figure 3A are more efficient at discovering (and installing) the coding table because they bypass long searches via natural selection. Thus, a translated peptide able to fold and which, when folded is able to catalyze aminoacyl-tRNA synthesis is analogous to a planetary orbit, freed from all the extraneous paths that don’t fulfill the gating criteria. Another way to say this is to recognize that the coding assignments and mRNA sequences of the aaRS genes search much smaller spaces because they are restricted to sequences that fold and folded proteins that enforce the coding rules. Thus, reciprocally-coupled gating makes them vastly more efficient and less vulnerable to toxic non-functional “parasites” that cripple autocatalytic sets. Coupled XNOR gates surmount Eigen’s paradox by this mechanism. Our argument parallels that of Deacon’s citation of Lao-Tse: “…use comes from what is not there” [129].

*Incompleteness and innovation*. In the horizontal dimension reciprocally-coupled gating achieves explicit self-reference, a property broadly linked to incompleteness because it facilitates construction of paradoxical sentences whose truth can neither be verified nor rejected within the axiomatic system in which they are constructed [20,131]. The statements associated with each of the examples described in §1(i)–(v) are inherently Gödelian sentences describing puzzles in Biology, each composing a similar “chicken and egg” paradox. Resolution of the paradoxes requires stepping outside the box, so to speak.

Although incompleteness is explicitly defined in Gödelian logic, and has never been proven to apply outside logic, we [9,11,102] have followed others [20,132,133,134,135,136,137] in highlighting that natural science may rhyme with logic. Most citations of incompleteness emphasize the negative, disappointing side of incompleteness—that logical systems cannot be complete. Our “cup half full” interpretation is shared by Dyson (“…because of Gödel’s theorem, physics is inexhaustible too.”) and Jaki (“…he (Hawking) made the erroneous claim that Gödel’s theorem means the end of physics” [132]. It means exactly the opposite.”) that incompleteness implies inexhaustibility. Inexhastibility, in turn, serves as a reservoir for innovation.

Completeness implies closure; it is self-limiting. Its opposite behaves like a high chemical potential of yet unformed novelty, hence it is a “teleodynamic” force [138,139]. The self-referential coupling of XNOR gates guarantees that the spiral superimposed on the two gates in Figure 8 is a source of novel "statements whose truth or falsehood cannot be determined” using the two if and only if criteria. A tangible example is a mutation that obeys both the folding rules and the coding rules, but which has an unexpected gain of function that Nature can exploit. A very large number of such "mutations" have never appeared, but could appear randomly at any time. Thus, within the spiral (to the left of the incompleteness arrow in Figure 8) there is a high concentration of these "virtual mutations" whereas fewer lie to the right of the arrow. That amounts to a concentration gradient which, by analogy to chemistry, amounts to forcing the diffusion of virtual mutations to move to the right, becoming real.

Incompleteness equips the strange loop in Figure 3B with the ability to function as a boot-block, i.e., to discover new functionalities not prefigured in the initial antecedent statements—in this case protein folding rules and aaRS codon sequences. The strange loop in Figure 7 behaves similarly, combining efficiency with discovery of more precise aaRS active sites and new coding letters.

The strange loop in Figure 4 is subtler with respect to its incompleteness. It suggests, however, a plausible model for the evolution of protein catalysts. Directed evolution of native and molten globular forms of dihydrofolate reductase improved catalysis of both forms using non-overlapping sets of mutants, without altering the basic structural differences between them [14]. The authors observed that selected mutations enhanced catalysis indirectly, by strengthening dynamic fluctuations that couple to the reaction coordinate. The strange loop in Figure 4A therefore may help reconcile an ongoing controversy over how conformational fluctuations contribute to catalysis [58,62,63,64,140,141] by resolving the apparent conflict between the pre-organizational basis of catalysis [58] and recurring evidence from multiple systems that dynamic networks can contribute to catalysis if transition state complementarity is a transient phenomenon [2].

*Broader applications*. Characterizing the dimensions of Figure 8 as “forces” leaves many details to fill in. Forces in physics arise from well-defined potentials and explicit gradients. They also are used to define explicit quantities such as work. Our rigorous derivations of the informational Ohm’s law in terms of errors and impedance matching [14] suggest that similar formalisms could be devised to bring forces associated with efficiency and incompleteness into better accordance with physics.

Even without formal validation, we believe the metaphorical interpretation of Figure 8 opens consequential new windows on the origin, function, and integration of biomolecules in biology. Remarkably, by establishing directionality in time, the implied forces clarify both mechanisms (example i) and origins (examples ii–v). We anticipate application to multiple additional phenomena. Efficient conversion of light energy into proton gradients by membrane proteins like bacteriorhodopsin [142]; transmembrane communication by G-protein coupled receptors [143]; and associated downstream regulatory circuits and intracellular regulation [144] all coordinate ligand binding to allosteric conformational changes. Thus, they appear to require some form of reciprocally coupled gating analogous to that summarized in Figure 2. Three distinct control mechanisms in the eukaryotic nucleus—the histone code [145,146], RNA splicing and mRNA assembly [147], and developmental control by HOX proteins [148]—have evolved via proliferation, coordination, and pruning of relatively low specificity interactions, from which Nature selected and elaborated the most useful (e.g., Figure 4 and Figure 7). Their evolution, and the mechanisms by which they enhance specificity of gene expression are vulnerable to the proliferation of dysfunctional variants analogous to parasitic viral sequences that cause mutational meltdown. These transcend bacterial regulatory mechanisms in ways that invite description as strange loops. It would be surprising not to identify reciprocally-coupled gates that fortify understanding of those processes as well.

The paradigm in Figure 8 may impact human health at a meta-level in as yet unanticipated ways. We only begin to understand the impact that mutational meltdown can have in designing antiviral strategies [149]. Defining appropriate reciprocally-coupled gating could potentially improved quantitative modeling to enhance such efforts. Moreover, most of the systems listed in the previous paragraph are central to pharmacologic intervention. Reciprocally-coupled gating may point to new insights that improve therapeutic regimens in such resistant problem areas as KRAS-linked oncogenesis [144].

## Figures and Tables

**Figure 1 biomolecules-11-00265-f001:**
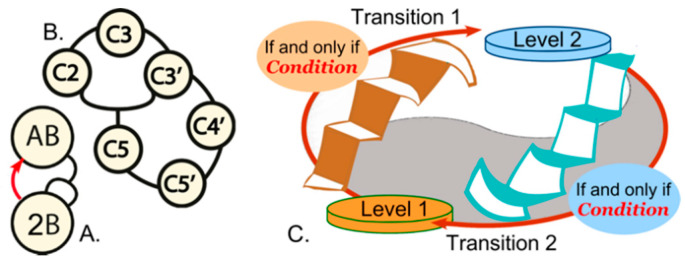
Biological process control blocks. (**A**). Simple autocatalysis. Reaction of molecules A and B gives rise to two B molecules, which therefore increase the rate at which B is produced. (**B**). A reflexive autocatalytic set, often called a Reflexive Autocatalytic Food Set (RAFS). The network is closed because the product of catalyst 5 is the substrate for catalyst 2 (reproduced from [19]). (**A**,**B**) were adapted from [19]. (**C**). Elements of reciprocally coupled gating, presented as a strange loop [20]. White stairs on left and right sides are filtered by an “if and only if” condition as both ascend from level to level and the antecedent of each staircase is the consequent of the other.

**Figure 2 biomolecules-11-00265-f002:**
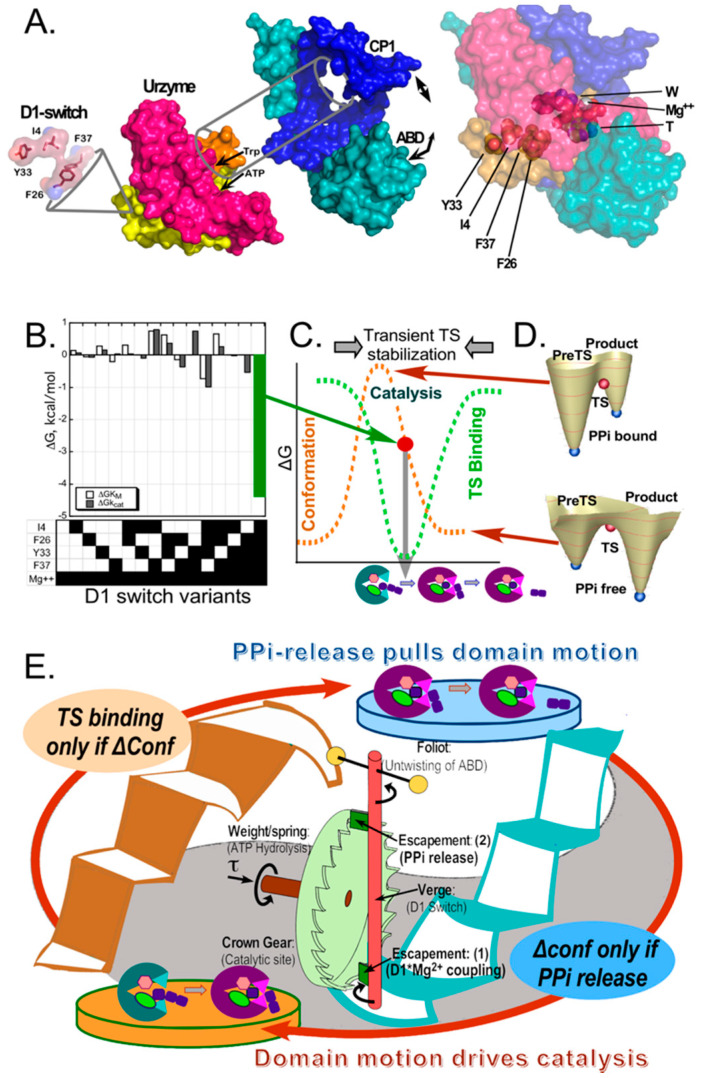
Reciprocally-coupled gating in the TrpRS escapement mechanism. (**A**). Structural biology of the TrpRS monomer. The catalytic machinery is located within the Urzyme and is modified by its interaction with two accessory domains, including the CP1 (Connecting Peptide 1 [67,68]; blue) and the anticodon-binding domain (ABD; teal), which move relative to the active-site. The D1 switch is located within the Protozyme (yellow), which is the first crossover connection of the Rossmann fold within the Urzyme. The specificity helix (sand) connects the first and second halves of the Urzyme (yellow, red). The D1 switch is~20 Å from the active-site metal ion. (adapted from [2].) (**B**). Combinatorial mutagenesis of the TrpRS D1 switch revealed that the entire catalytic contribution of the catalytic Mg^2+^ ion (~5 Kcal/mole) can be attributed to the 5-way interaction between it and the four D1 residues. (**C**). B–D were adapted, with permission, from [6]. Multiple studies [69,70,71,72] showed that the PreTS state TrpRS conformation is an excited state~3 kcal/mole higher in energy than either the open or Products ground states (brown dashed curve). Data from (**A**) show that the conformation complementary to the transition state for amino acid activation develops transiently, during domain motion from the PreTS to the Products conformation (green dashed curve). (**D**). Computational free energy surfaces of the transition between PreTS and Products states [4] with and without the product PPi show that the conformational equilibrium shifts to favor the Products conformation if and only if the PPi is absent from the active site. (**E**). Schematic representation of the reciprocally coupled gating as a strange loop, emphasizing the parallel between the data in (**B**–**D**) and the two green blades (Escapement (1) and (2)) that make the rotation of the crown gear in the escapement mechanism of a mechanical clock conditional on the rotation cycle of the pendulum or Foliot.

**Figure 3 biomolecules-11-00265-f003:**
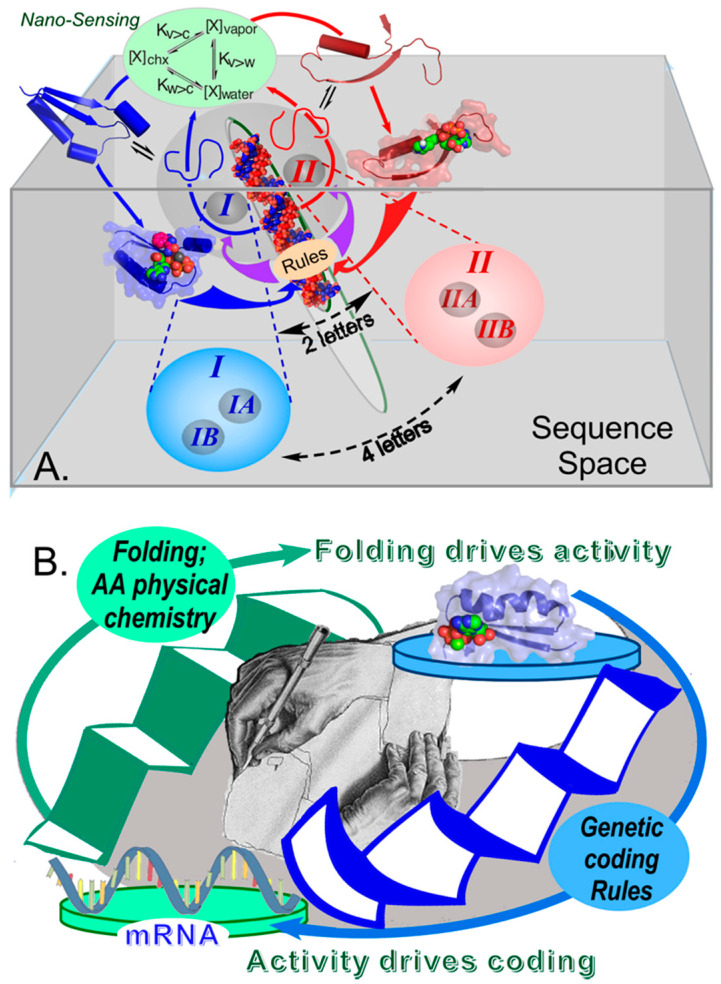
Reciprocally-coupled gating in the origin of genetic coding. (**A**). A single bidirectional gene encoding ancestral genes for Class I (blue) and Class II (red) aaRS on opposite strands combines with nanosensing arising from the equilibrium distributions of amino acid side chains between vapor, water, and cyclohexane [8,9,10,78,79,80,81] provide a boot-block for installing biology’s operating system by furnishing the minimal instruction set necessary to launch coded peptide-bond formation [11,12] based on two distinct amino acid types (adapted from [101]). (**B**). The underlying process control structure of that boot-block. Sequences are active if and only if they fold; and active sequences fold if and only if they obey the relevant coded sequence, which the aaRS collectively themselves enforce. The central graphic is adapted from M. C. Escher (adapted from [102]).

**Figure 4 biomolecules-11-00265-f004:**
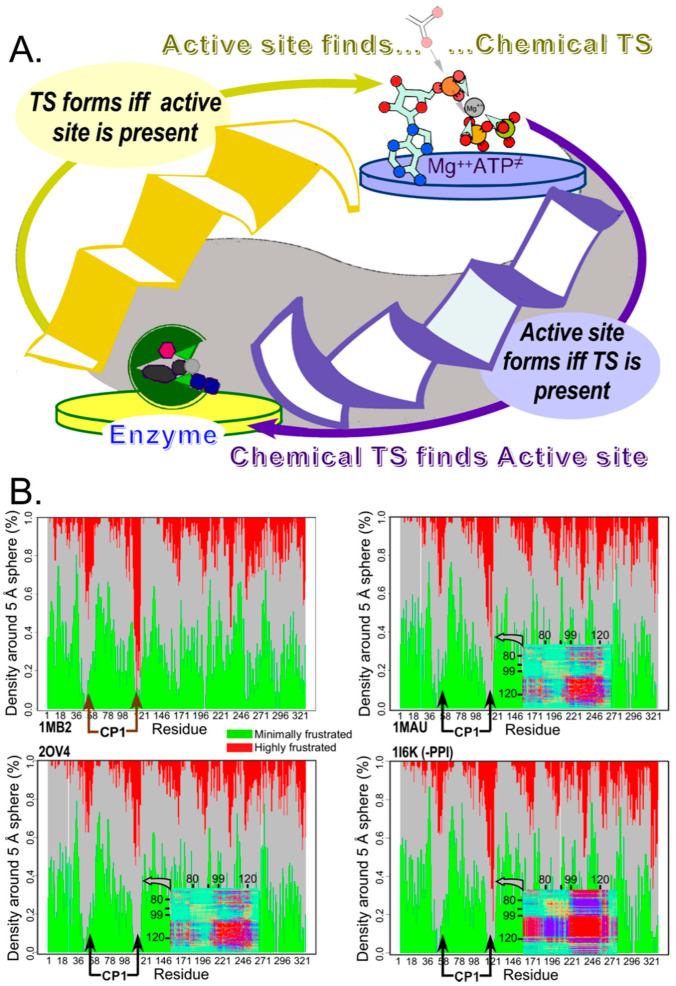
Reciprocally-coupled gating drove the evolution of catalysis. (**A**). The paradox of catalysis by molten globular ensembles is that both catalyst and reaction transition state are very rare species whose concentrations are too small to find one another often enough to accelerate the uncatalyzed rate. Yet the catalyzed rates of native and monomeric chorismate mutase are essentially the same, implying that the transition-state complex is as stable for the molten globular catalyst as for the native enzyme. The strange loop representing catalysis exhibits the same formal structure as those in Figure 1, Figure 2 and Figure 3, in which both catalysis and folding are governed by bi-conditional if and only if statements for which the antecedent and consequent states are interchanged. (**B**). Structure-based bioinformatics evidence suggests that even native TrpRS retains features of such coupling. TrpRS crystal structures of ground-state (1MB2), activated Pre-transition state (1MAU), transition state analog (2OV4), and Products (1I6K) state complexes exhibit a progression in which the regions where the mobile CP1 domain is inserted into the Urzyme maximally relax their severe frustration [108,109] in the transition-state complex. The inserts in the last three frustratograms are vignettes of the covariance heat maps obtained from replica exchange Discrete Molecular Dynamics simulations, and show that correlate motion at the *C*-terminal boundary between the Urzyme and CP1 intensifies only in the Products state and only after release of product, PPi.

**Figure 5 biomolecules-11-00265-f005:**
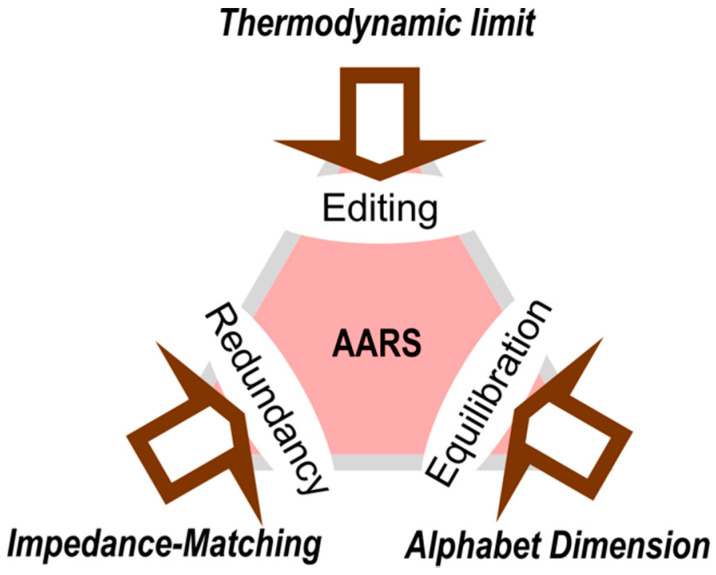
Simultaneous constraints on the precision of aminoacylation by aaRS. The alphabet size and fundamental thermodynamics both place different irreducible limits on aaRS precision. Any change in the dimension of the genetic coding alphabet must be followed by a round of equilibration, as new possibilities are introduced to optimize the catalytic and specificity of the extant aaRSs. Finally, minimizing the dissipation of information similarly constrains optimal aaRS precision according to the frequency of replication and transcription errors, in large part because of changes in coding redundancy [18].

**Figure 6 biomolecules-11-00265-f006:**
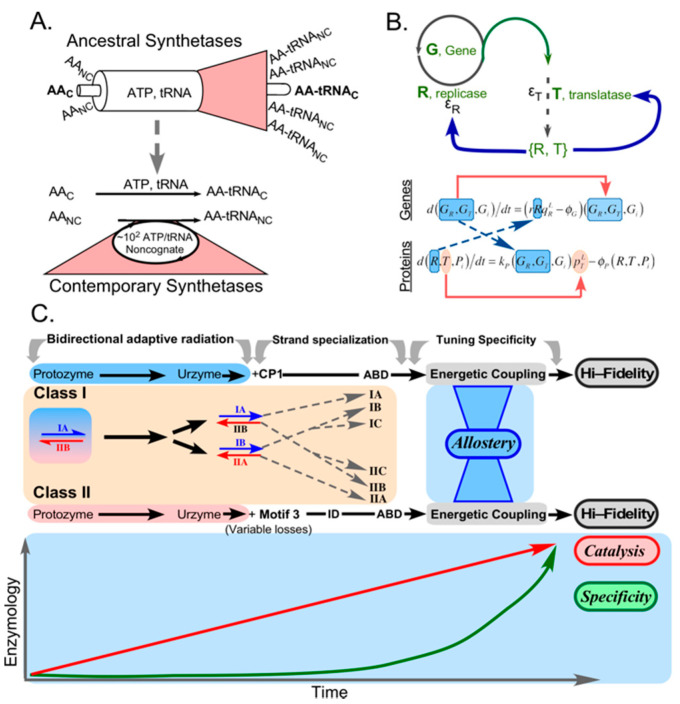
Challenges facing the evolution of aaRS associated with expanding the coding alphabet by introducing new aaRS tRNA cognate pairs. (**A**). High precision cannot be achieved using a single association dissociation equilibrium because some amino acids have similar side chains and bind to non-cognate aaRS with dissociation constants nearly 1% of those for cognate amino acids. Nature had to couple multiple binding events without allowing dissociation to enable contemporary aaRS to surpass this thermodynamic limitation. As a consequence, dissipation of information in mRNA codescripts due to translational errors by ancestral aaRS (horizontal red fan) was converted into dissipation of chemical free energy in contemporary aaRS (vertical red fan). (**B**). Coupling of replication to translation illustrated by the Gene-Replicase-Translatase model system. Rudiments of the GRT system are the replicase and translatase catalysts and their genes (green). Processes necessary to generate the active catalysts, R and T, are replication (circle), translation (dashed line), and folding (blue lines). No distinction is made between duplication and transcription of the respective genes, G, in a world where genetic information is instantiated in RNA. Errors are denoted by ε. Differential equations for the two processes [12] are coupled in both directions via the population variables (dashed arrows) and exhibit autocatalysis (red arrows). (Parts A and B were adapted from [18].) (**C**). Evolutionary events in aaRS evolution. (Top) The schematic illustrates three distinct stages in the structural and genetic development of aaRS. (Bottom) Experimental data [98] have shown that the catalytic proficiency of both classes is a linear function of the number of amino acids, and by implication with time. Specificity, however, likely failed to develop until the allosteric mechanisms could accomplish fine tuning.

**Figure 7 biomolecules-11-00265-f007:**
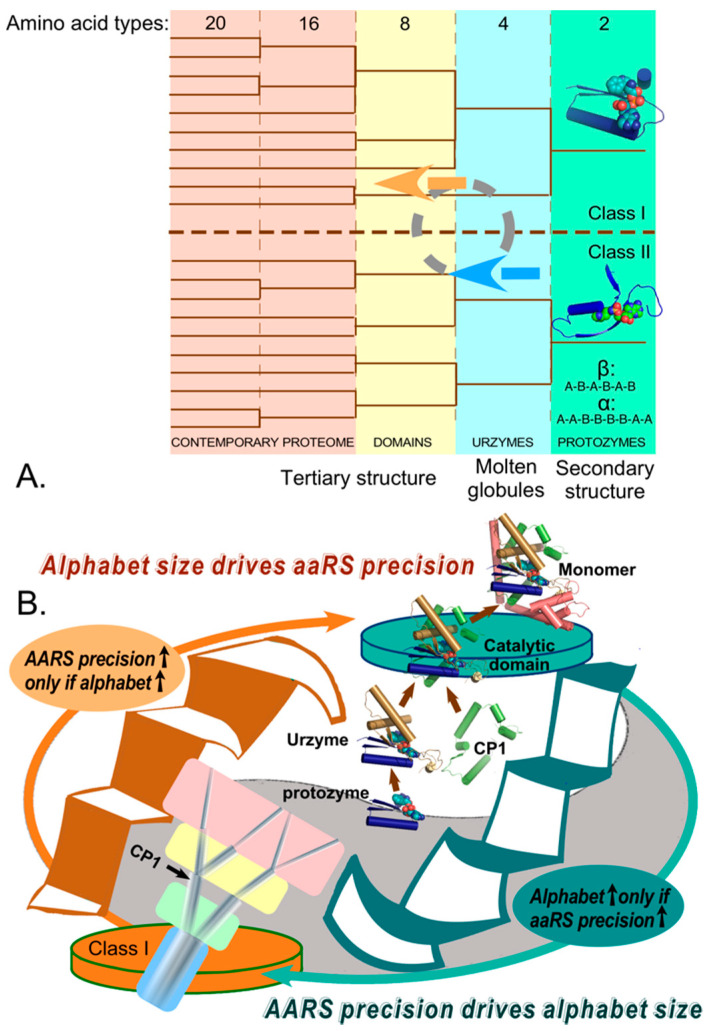
The strange loop connecting the proteome itself to the genetic coding table, via the precision of the aaRS and the coding alphabet size. (**A**). Stages in the evolution of the genetic coding alphabet (vertical panels; number of amino acid types) appear to be associated with the evolution of the proteome (bottom), beginning with the introduction of binary patterns enabling secondary structure formation (aaRS protozymes shown as cartoons) through intrinsically disordered molten globules (Urzymes), and finally unique tertiary structures [121]. Trees are represented by thin lines, and major increases in the size of the coding alphabet are represented in differently-colored panels. The circular symbol represents the remodeling of sequences that becomes necessary each time the number of distinct amino acid types increases. That sequence re-shuffling, optimizes in turn the precision of both new and extant aaRS genes, consistent with the new alphabet. (**B**). The reciprocally-coupled gating strange loop that drives increases in the alphabet size and aaRS precision.

**Figure 8 biomolecules-11-00265-f008:**
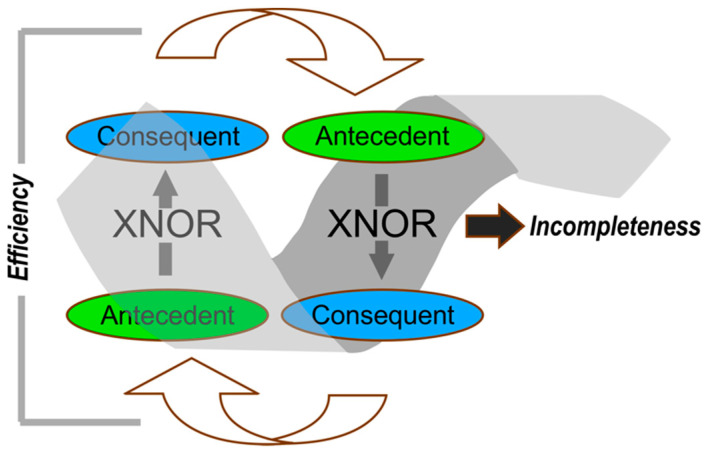
Elements of reciprocally-coupled gating. Two coupled logical XNOR gates are joined head to tail into a single gate, showing the coupling of antecedent and consequent “statements”, which assume different meanings in each of the examples described in Figure 2, Figure 3 and Figure 4 and Figure 7. That coupling is brought about by interchanging the antecedent and consequent. This type of coupling constitutes self-reference, hence creates incompleteness. The two XNOR filters together greatly damp unintended variants of the antecedent and consequent, hence reduce dissipation.

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
