# Peer review of "Reciprocally-Coupled Gating: Strange Loops in Bioenergetics, Genetics, and Catalysis"

_biomolecules, 2021, doi:10.3390/biom11020265_

Round 1

Reviewer 1 Report

This paper is what I would call scholarly speculation.  The idea proposed here of strange loops is a speculative unification across a few examples of a particular type of positive cooperativity that may have relevance to the origins of life.  In this arena, where definitive experimental proofs are in short supply, informed speculation is one of the few ways to move forward.  So, I am supportive of publishing ideas like these. And, this work is scholarly in the sense that it makes extensive connections across the literature.  I consider this a useful contribution for stimulating ideas in the field.

However, I have two concerns.  First, it uses too much jargon, and terms such as strange loops, urzymes, Eigen's Cliff, paradox of specificity, are not sufficiently clear.  It complains about error catastrophe, but needs to devote more space to spelling out what those problems are.  More detail in the exposition in general would allow in a broader readership than the narrow group it seems aimed at at the moment.

Second, this paper reads more like a first flash of inspiration -- admittedly a very good one -- but would benefit from more digestion or development of a specific example or quantitative calculation or model.  The main point here would be to be predictive, to state something unexpected that comes from this reasoning, so readers could reason with this logic too.

Reviewer 2 Report

This manuscript by Carter and Wills is one of the most scientifically interesting and intellectually stimulating works I have had the pleasure of reading in a long time.  The authors propose five compelling scenarios to illustrate how coupling underlies, in essence, the evolution of biological processes as we know them today.  What is so appealing is the way the authors bring together fundamental concepts and experimental observations to build something that is far greater than the sum of the parts of each "scenario".  This was of interest to me both with respect to general considerations of how catalysis develops as a viable biological process, and specifically with respect to the genetic code. I will openly admit to approaching the genetic code sections a little warily as someone who has read many such hypotheses over the years and generally found them to be a little underwhelming. That was not the case here; the notions of errors, cost, specificity and efficiency being harnessed in a coupled system to enable code expansion from 2 to 20 was compelling and ultimately very satisfying. I say satisfying as it was in complete agreement with what is known about other related experimental systems the authors did not mention directly. Examples include the directed evolution of orthogonal additions of unnatural amino acids to the genetic code, and the divergent evolution of the catalytic activities of synthetase fragments.

Overall this is a wonderful paper that I strongly recommend publishing "as is". The authors have done a superb job and their writing should not, in my opinion, be changed in any way. 

Reviewer 3 Report

There are many things I do not like about this paper. I am still struggling to figure out exactly what the paper is about and what kind of useful predictions can be made.

“biomolecular strange-loops provide a natural metaphor (line 22) around which to organize these data, linking biology to the physics of information, free (line 23) energy, and the second law of thermodynamics”. I cannot tell whether this is a strong idea or not.

I am not certain what this paper is about?

What is written about TrpRS-IC makes sense but is in no way surprising. Enzymes often uncouple ATP hydrolysis from some kind of power stroke or conformational change. Coupling reaction completion to pyrophosphate release is not a surprise or revelation.

Error-catastrophe is an evolved feature that evolved with the genetic code, because, prior to evolution of the genetic code, there is little penalty for chemical failures. Until there is a feature that can outcompete or destroy another feature, there is little problem with chemical dead ends. Before evolution of the genetic code, negative selections are weak. After evolution of the code, negative selections become much more lethal.

Nothing in biology is “if and only if”. Mistakes matter in biology. Really, it is a question of what mistakes are tolerated and why. It has been hypothesized that tRNA charging errors drove addition of new amino acids to the genetic code. Are the authors aware of this hypothesis?

 “It is nearly certain that the aaRS Class duality arose in the form of a bidirectional (line 234) ancestral gene encoding an ancestral Class I aaRS on one strand and an ancestral Class II (line 235) aaRS on the opposite strand”

This is not true. The Carter-Rodin-Ohno model is falsified in multiple papers, and the falsification is easily confirmed. Class IIA aaRS (GlyRS-IIA) are simple sequence homologs of class IA aaRS (ValRS-IA and IleRS-IA). All class II aaRS derive from GlyRS-IIA. All class I aaRS derive from (probably) ValRS-IA. Please check WP_0719072 vs. WP_0956521 and WP_0962041 vs. WP_0115001. The model of bi-directional transcription-translation for aaRS enzymes was never a likely model. There are very few bi-directional genes in modern biology.

Zn-binding was important in folding the first aaRS enzymes.  

The authors do not understand the relationship of class I and class II aaRS enzymes. Also, they appear to have a weak model for evolution of the genetic code. I guess their model might be pasted together by reading a number of their previous papers, but a clearer model should be presented here. Clear models have been published for evolution of the code. Clear models are not referred to here.

In this paper, there is no consideration of the tRNA anticodon and its reading on the ribosome in evolution of the code. Certainly, the code evolved around the tRNA anticodon. The genetic code is limited in size by the reading of the tRNA anticodon and wobbling. The genetic code is 64 codons in mRNA but effectively 32 assignments maximum in tRNA because of wobbling. Why do the authors not know this? Evolution of translation elongation factor EF-Tu suppressed wobbling at the 3rd anticodon position. This allows the code to expand beyond ~8 amino acids.

The tRNA anticodon evolved to have a register of 3—the idea of reading two at a time, before evolution of EF-Tu, is reasonable, but this is because of wobbling. The frame was always 3-nt because of the structure of the tRNA anticodon loop. On an ancient ribosome, before evolution of EF-Tu, 2 out of 3-nts were read for each tRNA. The authors should brush up on evolution of tRNA, because evolution of tRNA will inform their model.

The idea of aaRS enzymes evolving from molten globules is weak (the “urzyme” hypothesis). I guess it is possible that GlyRS-IIA initially evolved this way. Then, both class II and class I aaRS derived from GlyRS-IIA, as described in published work.  

Do the authors think that (beta-alpha)8 barrels (TIM barrels) and sheets (Rossmann folds) evolved from molten globules? Many evolutionary processes in earliest evolution involve multiple duplications of a single motif, i.e. beta-alpha-beta-alpha. Some proteins may have evolved via molten globules, but many proteins did not.

There is a war between amyloid plaque formation and orderly, compact globular protein shapes that continues today in human disease. In ancient evolution, random polymers often formed overly long beta sheets that generated amyloid plaques. This limits the success of molten globule evolution of proteins in the ancient world. Hence, the evolution of orderly sheets and barrels.  

Figure 4: substrates and cofactors help enzymes to fold. For instance, class I and class II aaRS might have initially folded around tRNAs bound on opposite faces. Zn-binding also informed initial class II and class I aaRS folding. Substrate and ATP cofactor can also help.

Why is the genetic code sectored? What is the source of degeneracy? What do the authors think is the maximal complexity of the genetic code? Why are there 6, 4, 3, 2, and 1 codon sectors? Why is serine split into two sectors in the code? The genetic code model presented makes little sense and appears to make few likely predictions.

Authors do not understand the structure, evolution and function of tRNA or the anticodon. This story is written in tRNA sequences.

The idea that aaRS enzymes evolved from molten globules is simply wrong. These were long, complex proteins from their first appearance. GlyRS-IIA is the root for all class II enzymes; GlyRS-IIA gave rise to class IA enzymes (probably ValRS-IA), which gave rise to all class I aaRS. Nothing else makes sense for aaRS evolution.

Editing in aaRS is a systematic problem. Left half of code edits (except ProRS-IIA in Archaea). Archaea are older than Bacteria. The authors should focus on archaeal systems, aaRS enzymes and tRNAs.  

Strongly suggest the authors read more broadly. For example (and references therein):

https://www.preprints.org/manuscript/202009.0162/v2

Round 2

Reviewer 3 Report

See attached file 
